# Medication, age and abstinence days associated with low semen quality: A cross-sectional study in more than 7000 men visiting the centre for reproductive medicine

P.G.J. ter Horst[1]*, M.A. Edens[2], D. den Besten-Bertholee[1], L.W. Mulder[1], M.H.J.M. Curfs[3]

1 Department of Clinical Pharmacy, Isala, Zwolle, The Netherlands, 2 Department of Innovation and Science, Isala, Zwolle, The Netherlands, 3 Department of Fertility,Isala, Zwolle, The Netherlands

* p.g.j.ter.horst@isala.nl

## Abstract

Medication can affect semen quality by decreasing ejaculate volume, sperm concentration, or decreased sperm motility and sperm function in general. We performed a retrospective explorative cross-sectional study on any medication use and semen quality in our fertility clinic in men older than 18 years with a recorded semen analysis. Men were categorized based on medication use, i.e., any type, and no medication use. Exclusion criteria were incomplete semen analysis, azoospermia, a semen analysis after vasectomy, and days of abstinence less than 2 or more than 7 days before semen collection. The primary outcome was the composite endpoint of low semen quality (LSQ) according to the WHO. In total 722 men with medication and 6716 men without medication were included in the study. At the ATC-7 level (individual drug ATC-code), univariate (borderline) statistically significant associations using a cut-off of p < 0.100 were found for metformin (A10BA02), metoprolol (C07AB02) and lisinopril (C09AA03), which remained significant after adjustment for age and abstinence days. Our study is limited by the fact that information regarding exposure to information, as information was self-reported by patients. Also, the outcome, semen quality, was hindered by patients own collection. Even if a drug had logically been used for some time before semen sampling (e.g., medication for chronic disease), there was no way to determine whether that medication had been started before or after the occurrence of LSQ. Finally, medication and disease cannot be separated, hence it could not be determined whether the medication or the disease itself was associated with LSQ. Therefore, this study should be interpreted as a hypothesis-generating study.

**Data availability statement:** "All relevant data are within the paper and its Supporting Information files."

**Funding:** The author(s) received no specific funding for this work.

**Competing interests:** The authors have declared that no competing interests exist.

## Introduction

Infertility is a health issue that affects approximately 15% of couples, who are at their reproductive age, and desiring to have children. [1] This condition is defined by the inability to achieve pregnancy after 1 year of regular unprotected sexual intercourse. [2] Approximately 50% of the infertility cases are attributed to males. [1] There is evidence that there is a decline in semen quality worldwide and particularly in Western countries which may be related to increased exposure to endocrine disruptors, immunological disorders, and obstruction of the reproductive tract. In addition, unhealthy lifestyles like excessive alcohol consumption and smoking, obesity, and the rising age of reproducing men are suggested. [3,4] Some of these factors also contribute to a decline in the overall health status in the last decades, and are causing an increase in the use of medication. Medication, prescribed and non-prescribed, may be associated with a decline in semen quality in men of reproductive age. In the Netherlands, the overall use of prescribed medication in the male population between the ages of 20–60 years is approximately 55%. [5–7] Research shows that decreased general health in men together with the use of medication in general may harm semen quality. [8]

Medication can influence semen quality through various mechanisms. Studies have shown that medication can impair hypothalamic-pituitary-gonadal functions, increase sperm DNA fragmentation and apoptosis, and reduce semen quality. [9] When medication interferes with the endocrine function of the testes by altering Leydig cells or disrupting the hormonal regulation, the drop in testosterone level can affect sperm production. [10] Other examples are theophylline which showed testicular atrophy in rats [11]; a single dose of paracetamol disrupted spermatogenesis at 10 days in rats [12]; morphine in rats resulted in cytogenetic abnormalities in spermatocytes which resolved after 13 weeks of drug withdrawal [13]; a single dose of aspirin disrupted the seminiferous tubule and when it was used for 35 days the total sperm count and motility were decreased. [12] Spironolactone, an aldosterone antagonist, is well known to cause various side effects such as loss of libido and impotence in males. [14]

Information about the safe use of medication related to spermatogenesis is scarce and clinical trials are lacking. Further, most information about medicines and semen quality is derived from animal studies rather than from humans. Therefore, this retrospective explorative cross-sectional study aims to investigate the use of medication and possible associations with low semen quality according to the WHO criteria [15] on sperm parameters compared to semen parameters in men without medication use.

## Methods

### Study design

We performed a retrospective explorative cross-sectional study on medication use and semen quality.

### Ethical approval

The Medical Ethical and Institutional Review Board of Isala gave a waiver for the study protocol on March 12th, 2020 (number 200311). Further, they waived the need for informed consent, as data were anonymized at time of final analysis.

## Setting

This study was conducted at the Fertility department of our clinic, a teaching hospital with a large affiliation area. We studied the semen quality of men from couples who visited the gynecology fertility department from 2007 to 2018 because of unfulfilled child wish. We accessed all records from the electronic patient files in March 2020 and were able to view identifiable data to retrieve all medical information on our subjects available in the files. We ended data collection on December the 1st 2020.

## Study population

All male partners of couples aged older than 18 years with a recorded semen analysis were eligible for inclusion. The indication for semen analysis was a visit to the fertility center because of the couples' infertility. Men were categorized based on medication use and no medication use (Fig 1). Exclusion criteria were incomplete semen analysis, azoospermia, a semen analysis after vasectomy, and days of abstinence less than 2 or more than 7 days before semen collection.

## Outcome, exposure, potential confounders

**Outcome.** The primary outcome was the composite endpoint of low semen quality, as defined by the WHO laboratory manual for the examination and processing of human semen, 5th edition. [15] This composite endpoint includes (1) ejaculate volume less than 1.5 mL, or (2) sperm concentration less than 15 million per mL, or (3) total sperm count less than 39 million, or (4) less than 32% progressively motility. If any of these was positive, "low semen quality" (LSQ) was diagnosed, meaning a lower chance of achieving a spontaneous pregnancy was expected.

**Semen analysis.** The following semen sample data were extracted: ejaculated volume (mL), sperm concentration ($10^6$/mL), sperm motility parameters (motile sperm, type A and B), non-progressive and percentage of immotile sperm, type C plus D), total sperm count ($10^6$/ejaculate), and total motile sperm count (TMSC, $10^6$/ejaculate) calculated as the

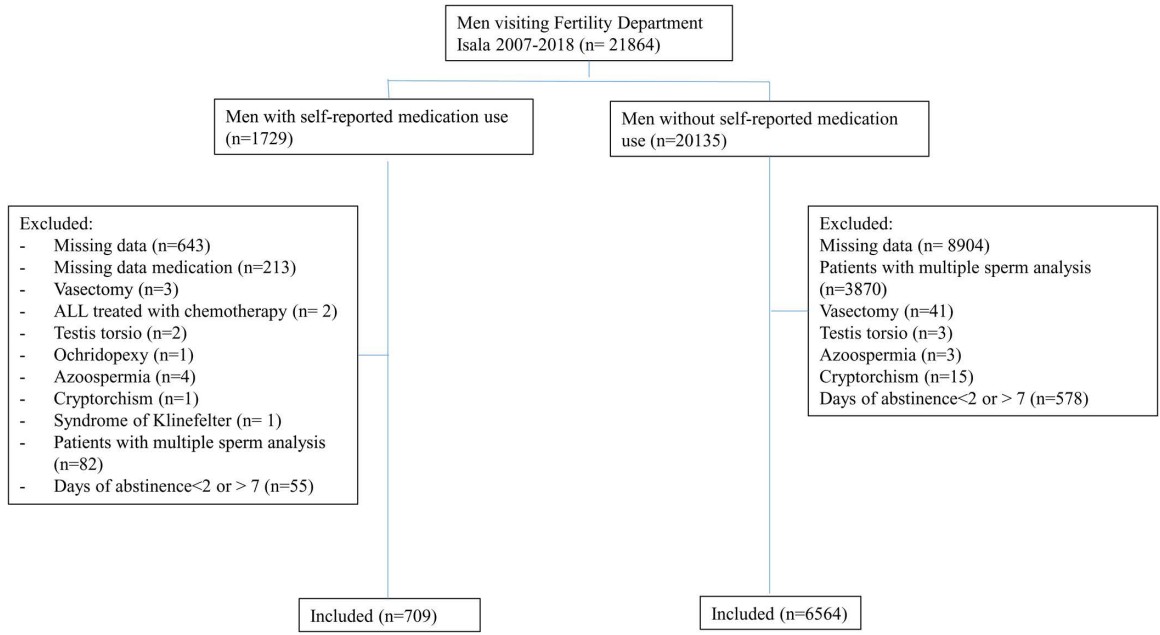

**Fig 1. Flowchart of included and excluded patients.**

product of ejaculate volume, sperm concentration, and grade A+B motility. [15] Expert laboratory staff of the fertility center at Isala performed all semen analyses.

**Medication use exposure.** Medication use of any type was our exposure of interest. The patient self-reported medication use during the time of semen collection was classified according to the Anatomical Therapeutic Chemical (ATC) classification system. Some medications were not specified and could not be classified according to the ATC classification system. Information on dosage and duration was not available and therefore not included. We divided medication at three levels: ATC at level 3 which comprises medication used for a disease (e.g., drugs used in diabetes); ATC 5 level which comprises a chemical subgroup (e.g., biguanides) and ATC at level 7, which comprises a specific chemical substance.

**Potential confounders.** We a-priori defined age as a possible confounder and categorized into the following age-groups: lower than 35 years old versus older than 35 years old; Lower than 35 years old versus older than 40 years, because of decreasing of semen quality when men got older.

Other potential confounders may be the time of abstinence of ejaculation (less than two or more than 7 days before semen collection).

**Data sources and measurement.** Data were extracted from electronic patient files of the fertility department.

**Patient and public involvement.** Patients and public were not involved in the design and conduct of this study. However, the results from this study may be helpful in deciding whether to use specific medication for men trying to inseminate.

## Proxy

Medication use was also used as a proxy for conditions/diseases. E.g. men using metformin were considered to have type 2 diabetes mellitus.

## Statistical methods

We used SPSS (IBM Corp. Released 2023. IBM SPSS Statistics for Windows, Version 29.0.2.0 Armonk, NY: IBM Corp) and STATA (StataCorp. 2023. Stata Statistical Software: Release 18. College Station, TX: StataCorp LLC) for all statistical analysis. We calculated LSQ prevalence with 95% Wilson confidence intervals, for: 1) all men included in the study, 2) those men on any type of medication(s), and 3) those men using specific medication at ATC code levels 3, 5, and 7.

Subsequently, we studied the univariate association of medication use with LSQ using the chi-square test, prevalence difference, prevalence ratio via Poisson regression with robust standard error [16,17], and prevalence odds ratio via logistic regression. [17,18]

Furthermore, we studied the discriminative value of variables for LSQ using receiver operating characteristic (ROC) analysis.

Confounding was considered by studying single medicament users and multivariable analyses. Medications borderline statistically significant (p<0.100) were further analyzed by multivariable analysis through Mantel-Haenzel (pooled), Poisson regression with robust standard error, and logistic regression analysis to enable adjustment.

## Results

### Subject characteristics

**Characteristics of the study population.** There were 21864 results of semen analysis, see Fig 1. From these analysis 722 men with medication and 6716 men without medication were included in the study, see Fig 1 Of all men with self-reported medication use, 54.4% used one kind medication, versus 45.6 using more than one medication. Baseline characteristics of the study group are presented in Table 1. Men with self-reported medication use were slightly older than

men without medication use (mean 34.7 ± 6.4 years and 33.7 ± 5.9 years, respectively). The duration of sexual abstinence was overall 2.8 days in both groups. The crude data for analysis are available as supplemental material.

**Low semen quality**

**Prevalence and association.** Fig 2a shows the LSQ prevalence for all included men and for men on medication(s). Of all n = 7273 included men, n = 3839 had LSQ, representing an overall LSQ prevalence of 52.8% (95%CI: 51.6% – 53.9%) which is indicated by the horizontal dotted line. Among the n = 709 men on medication (any type and any number), we found a higher LSQ prevalence of 56.1% (95%CI: 52.5% – 59.7%). Medication use (any type and any number) was borderline significantly associated with semen quality with p < 0.0599 (Fig 2b) which remained when adjusted for age and

**Table 1. Characteristics of the study group.**

| Variable | All (N = 7438) | Using Drugs (N = 722) | Not using drugs (N = 6716) | p-value |
|---|---|---|---|---|
| Age (y) | | | | <0.001 MW |
| Mean (sd) | 33.8 (±5.9) | 34.7 (±6.4) | 33.7 (±5.9) | |
| Median (Q1 – Q3) | 33 (30–37) | 34 (30–38) | 33 (30–37) | |
| Min – Max | 19–69 | 20–58 | 19–69 | |
| Year | | | | na |
| 2007 | 658 (8.8%) | 18 (2.5%) | 640 (9.5%) | |
| 2008 | 804 (10.8%) | 39 (5.4%) | 765 (11.4%) | |
| 2009 | 797 (10.7%) | 27 (3.7%) | 770 (11.5%) | |
| 2010 | 747 (10.0%) | 25 (3.5%) | 722 (10.8%) | |
| 2011 | 621 (8.3%) | 63 (8.7%) | 558 (8.3%) | |
| 2012 | 566 (7.6%) | 78 (10.8%) | 488 (7.3%) | |
| 2013 | 535 (7.2%) | 62 (8.6%) | 473 (7.0%) | |
| 2014 | 498 (6.7%) | 76 (10.5%) | 422 (6.3%) | |
| 2015 | 482 (6.5%) | 48 (6.6%) | 434 (6.5%) | |
| 2016 | 546 (7.3%) | 92 (12.7%) | 454 (6.8%) | |
| 2017 | 587 (7.9%) | 104 (14.4%) | 483 (7.2%) | |
| 2018 | 597 (8.0%) | 90 (12.5%) | 507 (7.5%) | |
| Days of abstinence | N = 7438 | 722 | 6716 | 0.129 MW |
| Mean | 2.8 (±1.0) | 2.8 (±1.0) | 2.8 (±1.0) | |
| Median (Q1 – Q3) | 2.5 (2–3) | 3.0 (2–3) | 3.0 (2–3) | |
| Min – Max | 2–7 | 2–7 | 2–7 | |
| Drug number | | N = 722 | | na |
| 1 | | 393 (54.4%) | | |
| 2 | | 196 (27.1%) | | |
| 3 | | 74 (10.2%) | | |
| 4 | | 40 (5.5%) | | |
| 5 | | 11 (1.5%) | | |
| 6 | | 6 (0.8%) | | |
| 7 | | 1 (0.1%) | | |
| 8 | | 1 (0.1%) | | |
| Not specified drugs | | 35 (0.5%) | 0 (0.0%) | na |

MW: Mann-Whitney U test; na: not applicable

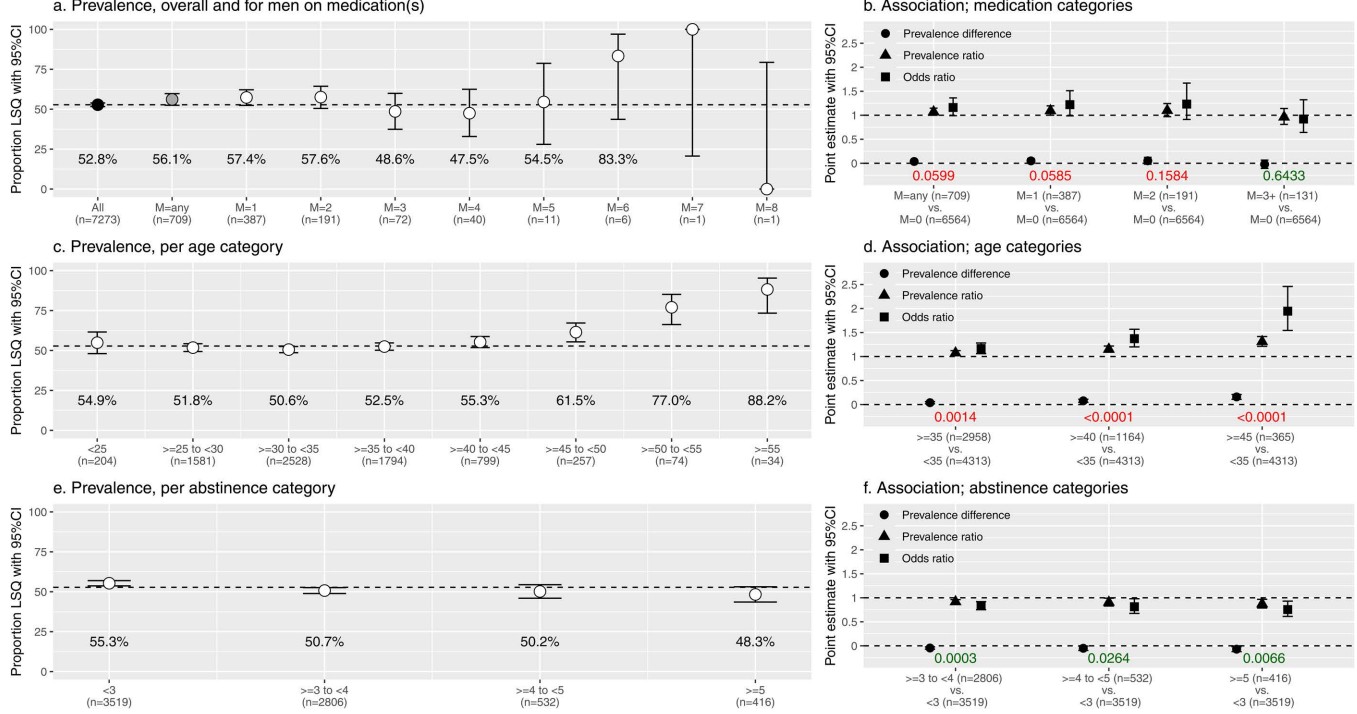

**Fig 2. A.** Proportion of men with Low Semen Quality (LSQ) including 95% confidence interval (CI) versus prevalence, overall and for men on medication(s) (M); B: Point estimates with 95% CI versus association with different medication categories; C: Proportion of men with Low Semen Quality (LSQ) including 95% CI versus prevalence, per age category; D: Point estimates with 95% CI versus association with different age categories; E: Proportion of men with LSQ including 95% CI versus prevalence, per abstinence category; F: Point estimates with 95% CI versus association with different abstinence categories.

abstinence days (Table 2). Furthermore, LSQ prevalence varied with the number of medications men were taking without a clear trend (Fig 2b).

Fig 2c shows LSQ prevalence per age category. In the lowest age categories up to 40 years, the LSQ prevalence was more or less around the overall LSQ prevalence of 52.8%. After 40 years, the proportion of LSQ gradually increased to 88.2% in men aged 55 years and older. In men aged 45 years and older, the LSQ prevalence ratio as compared to men younger than 35 was 1.3 with p < 0.0001 (Fig 2d), which shows a increasing negative association with LSQ as age increases. As – based on Fig 2c - LSQ prevalence does not increase up to 35 years, we chose <35 as reference category to which several classifications could be compared. We chose >=35, >=40, and>=45 based on methodological reasons. Higher age cut-offs would result in lower samples sizes decreasing reliability of results.

Figs 2e and 2f show an inverse association concerning prior semen sample abstinence days with LSQ. Point estimates show all significant negative associations in all categories (3–4 days vs < 3 days, p = 0.0003, 4–5 days vs < 3 days p = 0.0264, 5 days or more vs < 3 days, p = 0.066).

### Low semen quality per ATC code

**Prevalence and association.** LSQ prevalences with 95%CIs per ATC level are visualized in Figs 3, 4 and 5, and plotted against the overall prevalence of 52.8% indicated by the horizontal dotted line. LSQ prevalences were classified into 4 categories: 1] <52.8% (white), 2] 52.8% to <75% (light grey), 3] 75% to <100% (dark grey) and 4] 100% (black) as shown in Figs 3–5, and in Table 3.

**Table 2.** Univariable and multivariable analyses (age- and abstinence days-adjusted), concerning medicament(s) use and Low Semen Quality (LSQ).

| Medication | Univariate analysis | | | | |
|---|---|---|---|---|---|
| | p-value (Chi-square) | Prevalence difference | Prevalentie ratio/ Risk ratio Mantel-Haenszel pooled | Prevalence ratio IRR Poisson regression bin | Prevalence odds ratio Logistic regression |
| Medicament(s) use | 0.0599 | 3.7 (−0.1–7.6) | 1.071 (1.000–1.147) | 1.071 (0.999–1.147) bs | 1.161 (0.994–1.358) bs |
| Age | – | – | – | 1.009 (1.006–1.013) s | 1.020 (1.012–1.028) s |
| Abstinence days | – | – | – | 0.957 (0.934–0.981) s | 0.915 (0.873–0.960) s |
| Medicament(s) use | – | – | 1.057 (0.986–1.135) | 1.063 (0.993–1.139) bs | 1.146 (0.980–1.341) bs |
| Age | – | – | | 1.010 (1.006–1.013) s | 1.021 (1.013–1.029) s |
| Abstinence days | – | – | – – | 0.954 (0.931–0.978) s | 0.908 (0.865–0.952) s |

bs, borderline statistically significant (p < 0.10);

ns, not statistically significant (p ≥ 0.05);

s, statistically significant (p < 0.05).

In many cases LSQ prevalences were high, but sample sizes were too small to reach (borderline) statistical significance, also indicated by wide 95%CIs. In some cases, the combination of prevalence and sample size resulted in a (borderline) statistically significant association, indicated by p-values in red (positive associations) and by p-values in green (inverse association).

**Prevalence when using 1 medicament only.** Medicaments showing positive (borderline) statistically significant associations were selected for single medicament analysis. When selecting men that reported to use only one medicament, the findings remained largely intact. Some medicaments shifted to lower or higher classification (Table 3).

**Association.** The ATC codes that showed a (borderline) statistically significant association, and had at least n = 10 users, were selected for further statistical analysis (Table 4). At the ATC-7 level, univariate (borderline) statistically significant associations using cut-off p < 0.100 were found for metformin (A10BA02), metoprolol (C07AB02) and lisinopril (C09AA03).

When adjusted for age and abstinence days, these findings remained statistically significant, at least by Poisson analyses.

## Discussion

We found that metoprolol, metformin, and lisinopril were associated with a high LSQ prevalence. The numbers of medicament users were however not always considered sufficient for more advanced statistical analysis. Age and the number of abstinence days before semen sample collection were also associated with LSQ, independent of medication use. The study by Halvaei et al. (2020) showed that aging impairs sperm quality, which can affect the fertility status and the outcome of assisted reproductive technologies. [19]

In men visiting our fertility center and using one or more medications, semen quality may be impaired when compared to men without using medication. Especially, the use of metoprolol (beta-blocker), metformin (glucose-lowering drug) and lisinopril (ACE-inhibitor) seem to be associated with low semen quality, independent of age, however, with borderline significance at the p < 0.1 level.

We found a positive association between metoprolol and low semen quality, even when adjusted for age. The study by Guo et al. (2017) showed that the intake of beta-blockers resulted in a lower sperm concentration, motility, and total count. [20] One of the proposed mechanisms by which beta-blockers may induce sexual dysfunction and low semen quality is by inhibiting the sympathetic nervous system, which is involved in the integration of erection, emission, and ejaculation,

 

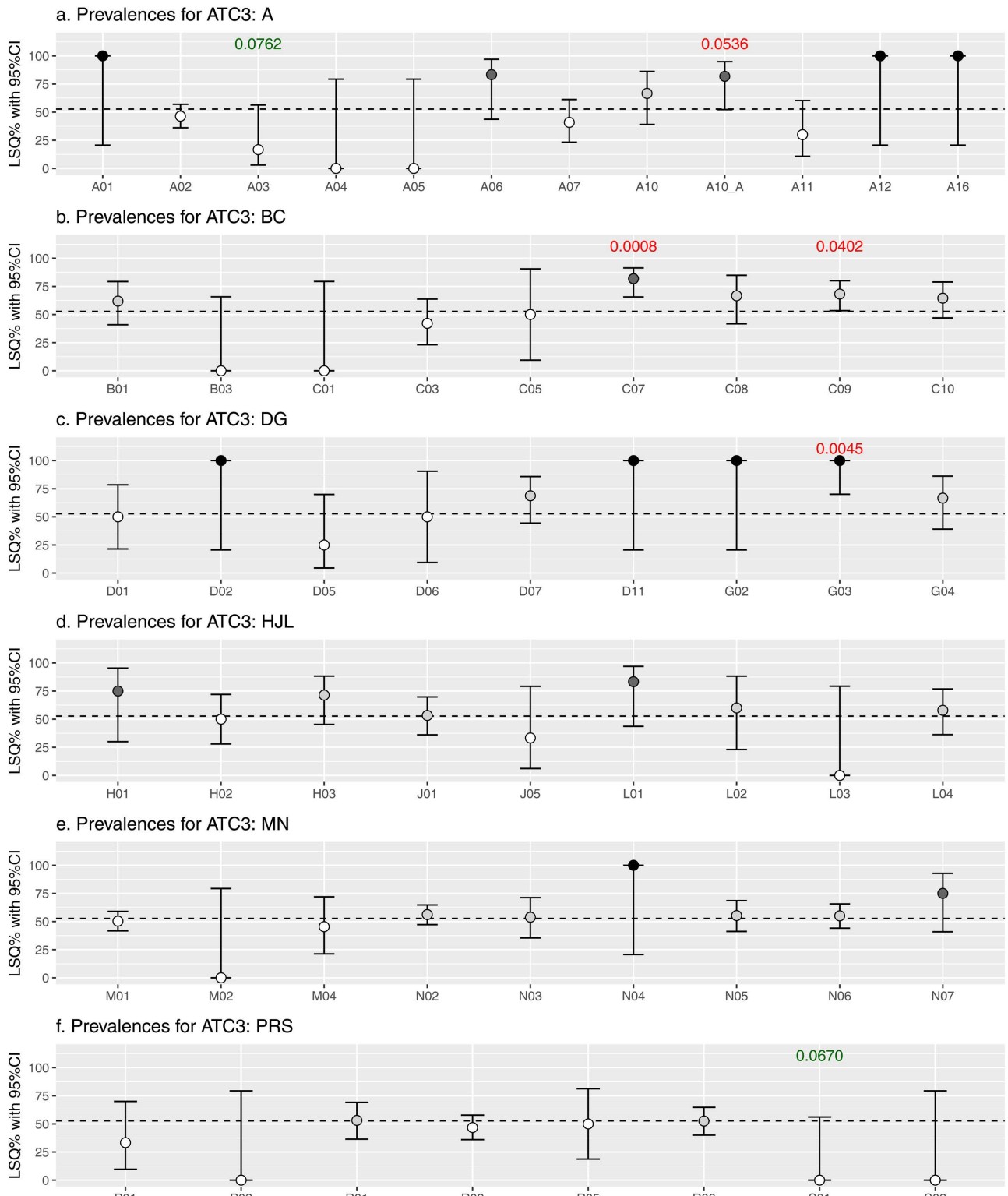

**Fig 3. Prevalence of low semen quality including 95% confidence interval (CI) for medication at the ATC 3 level.**

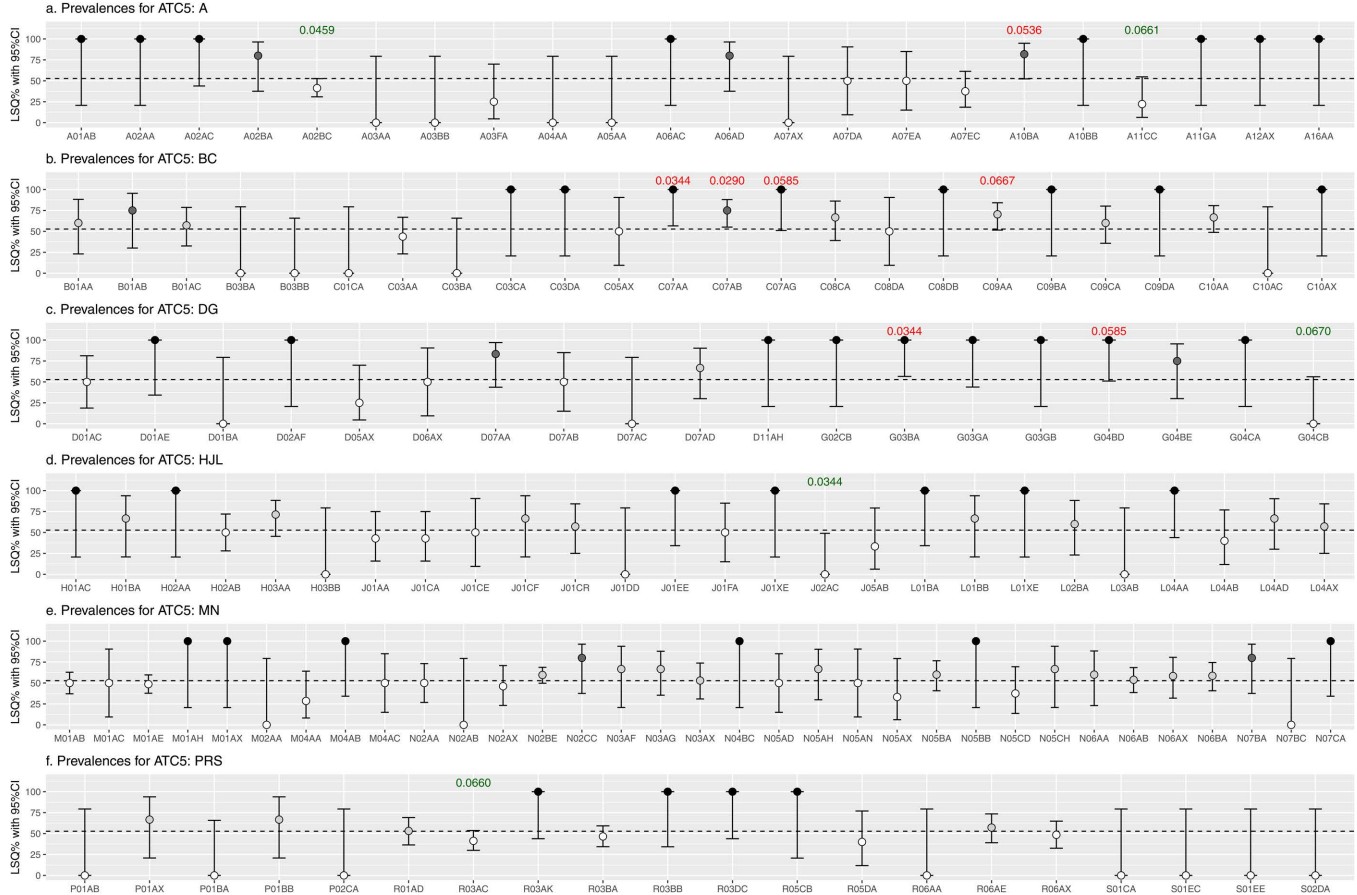

**Fig 4. Prevalence of low semen quality including 95% confidence interval (CI) for medication at the ATC 5 level.**

in the regulation of luteinizing-hormone secretion and the stimulation of release of testosterone. [21] A large cohort study by Skara et al (2022) found weak and no associations between infertility and cardiovascular disease (CVD) outcomes in men. This means a prescription of metoprolol is a proxy for CVD and therefor weakly or not associated with infertility. [21,22]

Arterial hypertension and hypercholesterolemia are considered risk factors for hormonal testicular function and spermatogenesis. [22] This risk is clinically relevant since hypertension affects up to 20% of the adult population. [23] The effects of agents acting on the renin-angiotensin system are not yet unequivocally demonstrated. A study by Schill et al. (1994) showed improved sperm concentration in men using captopril [24]; however, a retrospective study by Eisenberg et al. (2017) concluded that men taking beta-blockers or ACE inhibitors appeared to have a higher risk of infertility, which was not shown in men taking calcium channel blockers. [25] Agents acting on the renin-angiotensin system (enalapril, lisinopril, perindopril, ramipril, losartan, valsartan, candesartan, telmisartan) and sex hormones and modulators of the genital system (testosterone, human chorionic gonadotropin, human menopausal gonadotropin, and clomiphene) also seem to be associated with low semen quality, however, when adjusted for age, the association could no longer be demonstrated. A possible explanation is that older men more often use medication for the treatment of blood pressure control. [26] However, in this study, we could not demonstrate that the use of more than one agent was significantly associated with low semen quality. In particular, lisinopril in low dose (2.5 mg daily), was used in a randomized trial to study sperm parameters.

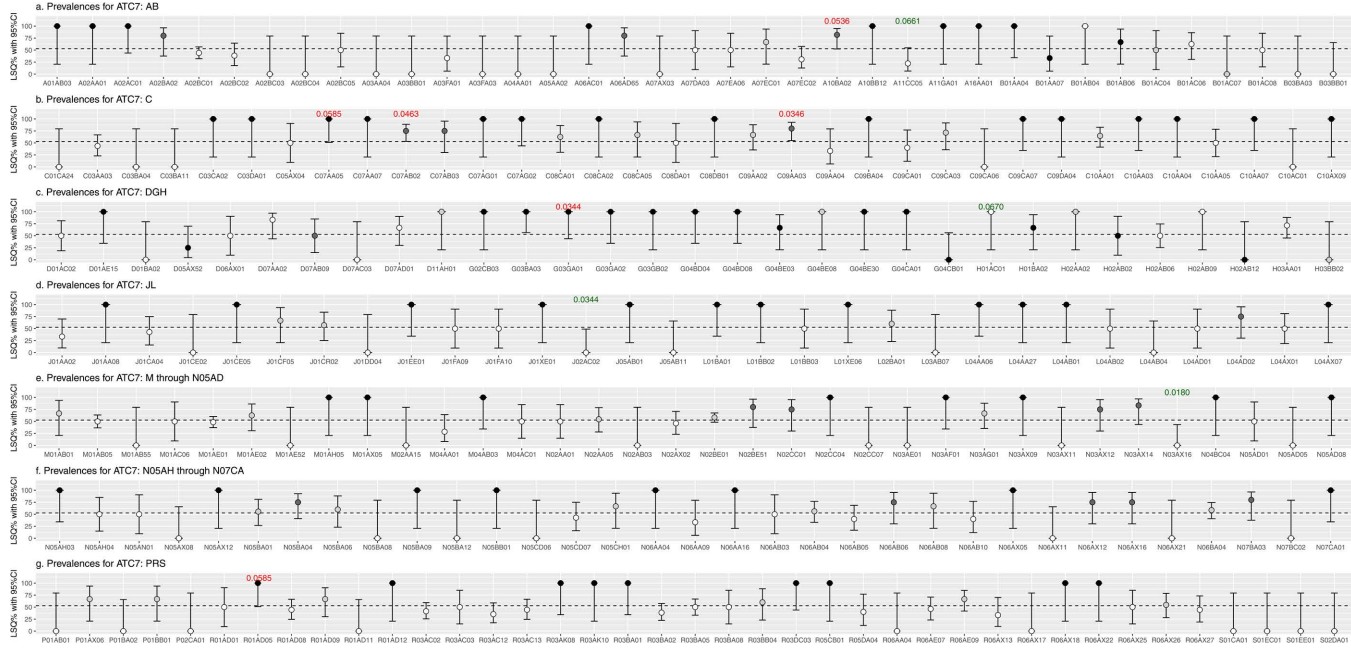

**Fig 5. Prevalence of low semen quality including 95% confidence interval (CI) for medication at the ATC 7 level.**

[27] Although the mean ejaculate volume was unchanged (P ≥ 0.093), the total sperm cell count and the percentage of motile sperm cells increased, whereas the percentage of sperm cells with abnormal morphology decreased. [27] Further studies are mandatory to explore the role of ACE inhibitors and semen quality.

Finally, we found metformin was associated with LSQ. A review published by Lotti and Maggi in 2022 showed that diabetes (type 1 and type 2) was associated with lower semen motility. [28] However, in our study it was hard to discriminate between medication effects (metformin) or (severity of) disease. Further studies are warranted because diabetes mellitus is one of the most common chronic conditions in Europe. At least 64 million adults and around 300 000 children and adolescents are estimated to be living with diabetes in the WHO European Region, see [https://www.who.int/europe/news-room/fact-sheets/item/diabetes]. [29]

Nifedipine is known for its disruptive influence on sperm motility. A study by Kanwar (1993) showed that the pattern of motility changed within two hours from rapid and linear progression to slow or sluggish linear or non-linear movement and finally to non-progressive motility or even immotility (in vitro). Scanning electron microscopic studies revealed disruptive changes in the head as well as tail region and coiling of spermatozoa after nifedipine treatment. [30]

Most information about the possible effects of medication on semen quality has been obtained from animal studies, and up to now, very little work has been performed to summarize the impact of medication on male fertility except for cancer treatments. A review by Semet et al. (2017) looked at the negative impact of pharmacological treatments on male fertility. Medication can impact male fertility by modification of the hypothalamic-pituitary-gonadal axis hormones or by non-hormonal mechanisms. [10] Some medication has a reversible effect on sperm parameters, like sulfasalazine. It has been observed that spermatogenesis generally recovers approximately 2–5 months after stopping sulfasalazine therapy. Testosterone supplementation therapy or anabolic steroids cause azoospermia or severe oligozoospermia. After the withdrawal of exogenous testosterone, spermatogenesis is spontaneously restored after approximately 3 months. Methotrexate, a cytotoxic agent that is also indicated for numerous autoimmune conditions, is gonadotoxic, mutagenic, and probably teratogenic, but its effects are reversed after the discontinuation of treatment. [10]

Table 3. Univariable and multivariable analyses (age-adjusted), concerning a selection of medicaments p < 0.1 and n ≥ 10.

| Medication | | | Univariate analysis | | | | |
|---|---|---|---|---|---|---|---|
| | | | p-value (Chi-square) | Prevalence difference | Risk ratio Mantel-Haenszel pooled | Prevalence ratio IRR Poisson regressie bin | Prevalence odds ratio Logistische regressie |
| Level | Type | n | | | | | |
| ATC 3 | A10_A crude | 11 | 0.0536 | 29.1 (6.3–51.9) | 1.551 (1.173–2.051) | 1.551 (1.173–2.052) s | 4.032 (0.871–18.676) bs |
| | A10_A age | 11 | | | 1.333 (1.017–1.749) – | 1.402 (1.057–1.858) s 1.009 (1.006–1.013) s | 3.255 (0.700–15.145) ns 1.020 (1.012–1.028) s |
| | C07 crude | 33 | 0.0008 | 29.2 (16.0–42.4) | 1.554 (1.321–1.828) | 1.554 (1.321–1.828) s | 4.047 (1.669–9.813) s |
| | C07 age | 33 | | | 1.401 (1.181–1.664) – | 1.462 (1.238–1.726) s 1.009 (1.005–1.012) s | 3.589 (1.475–8.731) s 1.019 (1.012–1.028) s |
| | C09 crude | 44 | 0.0402 | 15.5 (1.7–29.3) | 1.294 (1.056–1.585) | 1.294 (1.056–1.585) s | 1.924 (1.019–3.635) s |
| | C09 age | 44 | | | 1.145 (0.934–1.403) – | 1.215 (0.995–1.482) bs 1.009 (1.006–1.012) s | 1.689 (0.891–3.202) ns 1.019 (1.012–1.028) s |
| ATC 5 | A10BA crude | 11 | 0.0536 | 29.1 (6.3–51.9) | 1.551 (1.173–2.051) | 1.551 (1.173–2.052) s | 4.032 (0.871–18.676) bs |
| | A10BA age | 11 | | | 1.333 (1.017–1.749) – | 1.402 (1.057–1.858) s 1.009 (1.006–1.013) s | 3.255 (0.700–15.145) ns 1.020 (1.012–1.028) s |
| | C07AB crude | 24 | 0.0290 | 22.3 (4.9–39.7) | 1.423 (1.128–1.794) | 1.423 (1.128–1.794) s | 2.691 (1.067–6.788) s |
| | C07AB Age | 24 | | | 1.243 (0.974–1.586) – | 1.323 (1.047–1.672) s 1.009 (1.006–1.012) s | 2.331 (0.920–5.907) bs 1.020 (1.012–1.028) s |
| | C09AA crude | 27 | 0.0667 | 17.7 (0.4–34.9) | 1.335 (1.044–1.707) | 1.335 (1.044–1.707) s | 2.130 (0.931–4.872) bs |
| | C09AA age | 27 | | | 1.175 (0.922–1.499) – | 1.248 (0.983–1.583) bs 1.009 (1.006–1.013) s | 1.855 (0.808–4.261) ns 1.020 (1.012–1.028) s |
| | A02BC crude | 75 | 0.0459 | −11.6 (−22.8 – −0.4) | 0.781 (0.596–1.024) | 0.781 (0.596–1.024) bs | 0.627 (0.395–0.995) s |
| | A02BC age | 75 | | | 0.761 (0.581–0.998) – | 0.767 (0.586–1.003) bs 1.009 (1.006–1.013) s | 0.601 (0.378–0.955) s 1.021 (1.013–1.029) s |
| | R03AC crude | 63 | 0.0660 | −11.6 (−23.8–0.6) | 0.780 (0.581–1.049) | 0.780 (0.581–1.049) ns | 0.626 (0.378–1.036) bs |
| | R03AC age | 63 | | | 0.783 (0.582–1.053) – | 0.783 (0.583–1.052) ns 1.009 (1.006–1.013) s | 0.629 (0.380–1.042) bs 1.020 (1.012–1.028) s |
| ATC 7 | A10BA02 Metformine crude | 11 | 0.0536 | 29.1 (6.3–51.9) | 1.551 (1.173–2.051) | 1.551 (1.173–2.052) s | 4.032 (0.871–18.676) bs |
| | A10BA02Metformine age | 11 | | | 1.333 (1.017–1.749) – | 1.402 (1.057–1.858) s 1.009 (1.006–1.013) s | 3.255 (0.700–15.145) ns 1.020 (1.012–1.028) s |
| | C07AB02 Metoprolol crude | 20 | 0.0463 | 22.3 (3.3–41.3) | 1.423 (1.103–1.834) | 1.423 (1.103–1.834) s | 2.690 (0.977–7.409) bs |
| | C07AB02Metoprolol age | 20 | | | 1.289 (0.989–1.681) – | 1.332 (1.036–1.711) s 1.009 (1.006–1.013) s | 2.364 (0.854–6.542) bs 1.020 (1.012–1.028) s |
| | C09AA03 Lisinopril crude | 15 | 0.0346 | 27.3 (7.0–47.5) | 1.517 (1.177–1.956) | 1.517 (1.177–1.956) s | 3.586 (1.011–12.718) s |
| | C09AA03Lisinopril age | 15 | | | 1.275 (0.992–1.638) – | 1.409 (1.096–1.812) s 1.009 (1.006–1.013) s | 3.093 (0.868–11.018) bs 1.020 (1.012–1.028) s |

In this study we used the prevalence ratio's and odds ratio's, as measures of association based on prevalence of low semen quality. [31] The prevalence ratios may prevent the overestimation of odds ratios of the outcome as was demonstrated by Tamhane et al. (2016) and may be more accurate. [32]

**Table 4. Univariable and multivariable analyses (age- and abstinence days-adjusted), concerning a selection of medicaments with n ≥ 10 medicament users and p < 0.1 by univariate analysis.**

| Medication | | | Univariate analysis | | | | |
|---|---|---|---|---|---|---|---|
| | | | p-value (Chi-square) | Prevalence difference | Prevalentie ratio/ Risk ratio Mantel-Haenszel pooled | Prevalence ratio IRR Poisson regressie bin | Prevalence odds ratio Logistische regressie |
| Level | Type | N users | | | | | |
| ATC 3 | A10_A crude | 11 | 0.0536 | 29.1 (6.3–51.9) | 1.551 (1.173–2.051) | 1.551 (1.173–2.052) s | 4.032 (0.871–18.676) bs |
| | A10_A age | 11 | | | 1.333 (1.017–1.749) – | 1.402 (1.057–1.858) s 1.009 (1.006–1.013) s | 3.255 (0.700–15.145) ns 1.020 (1.012–1.028) s |
| | A10_A abstinence days | 11 | | | 1.569 (1.213–2.030) – | 1.555 (1.197–2.019) s 0.957 (0.934–0.981) s | 4.067 (0.877–18.863) bs 0.915 (0.873–0.960) s |
| | A10_A Age abstinence days | 11 | | | 1.300 (1.070–1.581) – – | 1.399 (1.076–1.818) s 1.010 (1.006–1.013) s 0.954 (0.931–0.978) s | 3.246 (0.697–15.115) ns 1.021 (1.01–1.029) s 0.908 (0.866–0.953) s |
| | C07 crude | 33 | 0.0008 | 29.2 (16.0–42.4) | 1.554 (1.321–1.828) | 1.554 (1.321–1.828) s | 4.047 (1.669–9.813) s |
| | C07 age | 33 | | | 1.401 (1.181–1.664) – | 1.462 (1.238–1.726) s 1.009 (1.005–1.012) s | 3.589 (1.475–8.731) s 1.019 (1.012–1.028) s |
| | C07 abstinence days | 33 | | | 1.602 (1.360–1.887) – | 1.578 (1.337–1.862) s 0.956 (0.933–0.980) s | 4.189 (1.726–10.169) s 0.913 (0.871–0.958) s |
| | C07 age abstinence days | 33 | | | 1.427 (1.210–1.683) – – | 1.480 (1.249–1.754) s 1.009 (1.006–1.013) s 0.953 (0.930–0.977) s | 3.714 (1.524–9.053) s 1.020 (1.012–1.029) s 0.907 (0.864–0.951) s |
| | C09 crude | 44 | 0.0402 | 15.5 (1.7–29.3) | 1.294 (1.056–1.585) | 1.294 (1.056–1.585) s | 1.924 (1.019–3.635) s |
| | C09 age | 44 | | | 1.145 (0.934–1.403) – | 1.215 (0.995–1.482) bs 1.009 (1.006–1.012) s | 1.689 (0.891–3.202) ns 1.019 (1.012–1.028) s |
| | C09 abstinence days | 44 | | | 1.307 (1.068–1.600) – | 1.302 (1.062–1.596) s 0.957 (0.934–0.981) s | 1.951 (1.032–3.687) s 0.915 (0.872–0.959) s |
| | C09 Age abstinence days | 44 | | | 1.124 (0.898–1.407) – – | 1.218 (0.998–1.487) bs 1.009 (1.006–1.013) s 0.954 (0.931–0.978) s | 1.704 (0.898–3.235) ns 1.021 (1.013–1.029) s 0.908 (0.866–0.952) s |
| ATC 5 | A10BA crude | 11 | 0.0536 | 29.1 (6.3–51.9) | 1.551 (1.173–2.051) | 1.551 (1.173–2.052) s | 4.032 (0.871–18.676) bs |
| | A10BA age | 11 | | | 1.333 (1.017–1.749) – | 1.402 (1.057–1.858) s 1.009 (1.006–1.013) s | 3.255 (0.700–15.145) ns 1.020 (1.012–1.028) s |
| | A10BA abstinence days | 11 | | | 1.569 (1.213–2.030) – | 1.555 (1.197–2.019) s 0.957 (0.934–0.981) s | 4.067 (0.877–18.863) bs 0.915 (0.873–0.960) s |
| | A10BA Age abstinence days | 11 | | | 1.300 (1.070–1.581) – – | 1.399 (1.076–1.818) s 1.010 (1.006–1.013) s 0.954 (0.931–0.978) s | 3.246 (0.697–15.115) ns 1.021 (1.013–1.029) s 0.908 (0.866–0.953) s |
| | C07AB crude | 24 | 0.0290 | 22.3 (4.9–39.7) | 1.423 (1.128–1.794) | 1.423 (1.128–1.794) s | 2.691 (1.067–6.788) s |
| | C07AB Age | 24 | | | 1.243 (0.974–1.586) – | 1.323 (1.047–1.672) s 1.009 (1.006–1.012) s | 2.331 (0.920–5.907) bs 1.020 (1.012–1.028) s |
| | C07AB abstinence days | 24 | | | 1.474 (1.166–1.864) – | 1.439 (1.138–1.819) s 0.957 (0.934–0.981) s | 2.752 (1.091–6.943) s 0.915 (0.872–0.959) s |
| | C07AB Age abstinence days | 24 | | | 1.272 (1.007–1.606) – – | 1.334 (1.053–1.690) s 1.009 (1.006–1.013) s 0.954 (0.931–0.978) s | 2.371 (0.935–6.013) bs 1.021 (1.013–1.029) s 0.908 (0.865–0.952) s |
| | C09AA crude | 27 | 0.0667 | 17.7 (0.4–34.9) | 1.335 (1.044–1.707) | 1.335 (1.044–1.707) s | 2.130 (0.931–4.872) bs |
| | C09AA age | 27 | | | 1.175 (0.922–1.499) – | 1.248 (0.983–1.583) bs 1.009 (1.006–1.013) s | 1.855 (0.808–4.261) ns 1.020 (1.012–1.028) s |
| | C09AA abstinence days | 27 | | | 1.340 (1.050–1.709) – | 1.342 (1.049–1.716) s 0.957 (0.934–0.981) s | 2.157 (0.942–4.937) bs 0.915 (0.873–0.960) s |

*(Continued)*

| Medication | | | Univariate analysis | | | | |
|---|---|---|---|---|---|---|---|
| | | | p-value (Chi-square) | Prevalence difference | Prevalentie ratio/ Risk ratio Mantel-Haenszel pooled | Prevalence ratio IRR Poisson regressie bin | Prevalence odds ratio Logistische regressie |
| Level | Type | N users | | | | | |
| | C09AA Age abstinence days | 27 | | | 1.172 (0.898–1.531) – – | 1.250 (0.985–1.586) bs 1.010 (1.006–1.013) s 0.954 (0.931–0.978) s | 1.870 (0.813–4.302) ns 1.021 (1.013–1.029) s 0.908 (0.866–0.953) s |
| | A02BC crude | 75 | 0.0459 | −11.6 (−22.8 – −0.4) | 0.781 (0.596–1.024) | 0.781 (0.596–1.024) bs | 0.627 (0.395–0.995) s |
| | A02BC age | 75 | | | 0.761 (0.581–0.998) – | 0.767 (0.586–1.003) bs 1.009 (1.006–1.013) s | 0.601 (0.378–0.955) s 1.021 (1.013–1.029) s |
| | A02BC abstinence days | 75 | | | 0.778 (0.594–1.019) – | 0.781 (0.596–1.024) bs 0.957 (0.934–0.981) s | 0.627 (0.395–0.995) s 0.915 (0.873–0.960) s |
| | A02BC Age abstinence days | 75 | | | 0.744 (0.567–0.976) – – | 0.766 (0.586–1.002) bs 1.010 (1.006–1.013) s 0.954 (0.931–0.978) s | ==0.599 (0.377–0.953) s== 1.022 (1.014–1.030) s 0.908 (0.866–0.953) s |
| | R03AC crude | 63 | 0.0660 | −11.6 (−23.8–0.6) | 0.780 (0.581–1.049) | 0.780 (0.581–1.049) ns | 0.626 (0.378–1.036) bs |
| | R03AC age | 63 | | | 0.783 (0.582–1.053) – | 0.783 (0.583–1.052) ns 1.009 (1.006–1.013) s | 0.629 (0.380–1.042) bs 1.020 (1.012–1.028) s |
| | R03AC abstinence days | 63 | | | 0.773 (0.575–1.038) – | 0.779 (0.579–1.047) bs 0.957 (0.934–0.981) s | 0.623 (0.376–1.031) bs 0.915 (0.873–0.960) s |
| | R03AC Age abstinence days | 63 | | | 0.770 (0.571–1.038) – – | 0.781 (0.580–1.050) ns 1.010 (1.006–1.013) s 0.954 (0.931–0.978) s | 0.625 (0.377–1.036) bs 1.021 (1.013–1.029) s 0.908 (0.866–0.952) s |
| ATC 7 | A10BA02 Met-formine crude | 11 | 0.0536 | 29.1 (6.3–51.9) | 1.551 (1.173–2.051) | 1.551 (1.173–2.052) s | 4.032 (0.871–18.676) bs |
| | A10BA02Metformine age | 11 | | | 1.333 (1.017–1.749) – | 1.402 (1.057–1.858) s 1.009 (1.006–1.013) s | 3.255 (0.700–15.145) ns 1.020 (1.012–1.028) s |
| | A10BA02Metformine abstinence days | 11 | | | 1.569 (1.213–2.030) – | 1.555 (1.197–2.019) s 0.957 (0.934–0.981) s | 4.067 (0.877–18.863) bs 0.915 (0.873–0.960) s |
| | A10BA02Metformine Age abstinence days | 11 | | | 1.300 (1.070–1.581) – – | ==1.399 (1.076–1.818) s== 1.010 (1.006–1.013) s 0.954 (0.931–0.978) s | 3.246 (0.697–15.115) ns 1.021 (1.013–1.029) s 0.908 (0.866–0.953) s |
| | C07AB02 Metopro-lol crude | 20 | 0.0463 | 22.3 (3.3–41.3) | 1.423 (1.103–1.834) | 1.423 (1.103–1.834) s | 2.690 (0.977–7.409) bs |
| | C07AB02Metoprolol age | 20 | | | 1.289 (0.989–1.681) – | 1.332 (1.036–1.711) s 1.009 (1.006–1.013) s | 2.364 (0.854–6.542) bs 1.020 (1.012–1.028) s |
| | C07AB02Metoprolol abstinence days | 20 | | | 1.473 (1.140–1.904) – | 1.440 (1.113–1.863) s 0.957 (0.934–0.981) s | 2.759 (1.001–7.603) bs 0.915 (0.872–0.959) s |
| | C07AB02Metoprolol Age abstinence days | 20 | | | 1.305 (1.005–1.695) – – | ==1.345 (1.043–1.734) s== 1.010 (1.006–1.013) s 0.954 (0.931–0.978) s | 2.414 (0.872–6.684) bs 1.021 (1.013–1.029) s 0.908 (0.865–0.952) s |
| | C09AA03 Lisinopril crude | 15 | 0.0346 | 27.3 (7.0–47.5) | 1.517 (1.177–1.956) | 1.517 (1.177–1.956) s | 3.586 (1.011–12.718) s |
| | C09AA03Lisinopril age | 15 | | | 1.275 (0.992–1.638) – | 1.409 (1.096–1.812) s 1.009 (1.006–1.013) s | 3.093 (0.868–11.018) bs 1.020 (1.012–1.028) s |
| | C09AA03Lisinopril abstinence days | 15 | | | 1.561 (1.211–2.012) – | 1.548 (1.194–2.008) s 0.957 (0.934–0.981) s | 3.750 (1.056–13.321) s 0.914 (0.872–0.959) s |
| | C09AA03Lisinopril Age abstinence days | 15 | | | 1.321 (0.981–1.779) – – | 1.433 (1.107–1.856) s 1.009 (1.006–1.013) s 0.954 (0.931–0.977) s | 3.232 (0.905–11.543) bs 1.021 (1.013–1.029) s 0.907 (0.865–0.952) s |

bs, borderline statistically significant (p < 0.10);

ns, not statistically significant (p ≥ 0.05);

s, statistically significant (p < 0.05).

Our study is limited, inherent to the study design, by the fact that information regarding exposure to information, as information was self-reported by patients. Also, the outcome, semen quality, was hindered by patients own collection. Therefore, caution is needed in concluding. Even if a medicament had logically been used for some time before semen sampling (e.g., medication for chronic disease), there was no way to determine whether that medication had been started before or after the occurrence of LSQ. Therefore, this study should be interpreted as a hypothesis-generating study. Finally, medication and disease cannot be separated, hence it could not be determined whether medication or the disease itself was associated with LSQ.

## Conclusion

We found in this hypothesis generating study that the use of metoprolol (beta-blocker), metformin (glucose-lowering drug) and lisinopril (ACE-inhibitor) seem to be associated with low semen quality, independent of age.

### Key Points

- Semen quality of 722 men using medication were analyzed and compared with the semen quality of 6716 men without using medication

- Medication might negatively affect semen quality

- In this hypothesis-generating study, metformin, metoprolol and lisinopril were associated with low semen quality

## Supporting information

**S1 Data.**
(CSV)

## Author contributions

**Conceptualization:** Peter ter Horst.

**Data curation:** L.W. Mulder, M.H.J.M. Curfs.

**Formal analysis:** Peter ter Horst, M.A. Edens.

**Investigation:** Peter ter Horst, M.A. Edens, M.H.J.M. Curfs.

**Methodology:** Peter ter Horst, M.A. Edens, L.W. Mulder, M.H.J.M. Curfs.

**Project administration:** Peter ter Horst.

**Resources:** Peter ter Horst.

**Software:** Peter ter Horst.

**Supervision:** Peter ter Horst, M.A. Edens, M.H.J.M. Curfs.

**Validation:** Peter ter Horst, M.A. Edens.

**Writing – original draft:** Peter ter Horst, M.A. Edens, D. den Besten-Bertholee, L.W. Mulder, M.H.J.M. Curfs.

**Writing – review & editing:** M.A. Edens, D. den Besten-Bertholee, L.W. Mulder, M.H.J.M. Curfs.

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
