## [Decision Letter · Decision Letter 0]

Thank you for submitting your manuscript to PLOS ONE. After careful consideration, we feel that it has merit but does not fully meet PLOS ONE’s publication criteria as it currently stands. Therefore, we invite you to submit a revised version of the manuscript that addresses the points raised during the review process.

If applicable, we recommend that you deposit your laboratory protocols in protocols.io to enhance the reproducibility of your results. Protocols.io assigns your protocol its own identifier (DOI) so that it can be cited independently in the future. For instructions see: https://://journals.plos.org/plosone/s/submission-guidelines#loc-laboratory-protocols . Additionally, PLOS ONE offers an option for publishing peer-reviewed Lab Protocol articles, which describe protocols hosted on protocols.io. Read more information on sharing protocols at https://://plos.org/protocols?utm_medium=editorial-email&utm_source=authorletters&utm_campaign=protocols .

We look forward to receiving your revised manuscript.

Kind regards,

Mukhtiar Baig, Ph.D.

Academic Editor

PLOS ONE

Journal Requirements:

https://://journals.plos.org/plosone/s/file?id=wjVg/PLOSOne_formatting_sample_main_body.pdf and 

https://://journals.plos.org/plosone/s/file?id=ba62/PLOSOne_formatting_sample_title_authors_affiliations.pdf

2. In the online submission form, you indicated that [The data underlying the results presented in the study are available from Isala (corrsponding author) upon reasonable request]. 

Reviewers' comments:

Reviewer's Responses to Questions

**Comments to the Author**

1. Is the manuscript technically sound, and do the data support the conclusions?

Reviewer #1: Yes

Reviewer #2: Partly

Reviewer #3: Yes

2. Has the statistical analysis been performed appropriately and rigorously?

Reviewer #1: I Don't Know

Reviewer #2: Yes

Reviewer #3: I Don't Know

3. Have the authors made all data underlying the findings in their manuscript fully available?

Reviewer #1: No

Reviewer #2: No

Reviewer #3: Yes

4. Is the manuscript presented in an intelligible fashion and written in standard English?

Reviewer #1: Yes

Reviewer #2: No

Reviewer #3: Yes

Reviewer #1: Well written. However, it is a Retrospective study. As required by the journal, you need to have the data available. The graphs are also not easy to understand. Otherwise discussion and references were acceptable.

Reviewer #2: In this study, the authors investigated the impact of medication on semen quality as per WHO criteria in a total of 7438 patients, of which 722 patients were under medication at the time of semen analysis, at Isala Fertility Center. Men were categorized based on the type of medication used and no medication. Age and period of sexual abstinence were investigated as confounding factors. Based on the medication used (the authors highlight metformin, metoprolol and lisinopril), the relevant medical conditions were concluded. Results show some significant effects of the medication taken on semen quality independent from age.

Although this is a very important aspect in andrological diagnostics for which the authors should be commended, the manuscript has significant problems starting from the English which has to be improved. In addition, important explanations are missing (e.g. ATC-7 level). The medical conditions are not mentioned in the Materials and Methods section. The tables that the authors are referring to are missing. The Figures do not have a legend. Furthermore, the authors link poor semen quality with intake of metformin. However, since metformin is used to treat diabetes, the impact of diabetes on semen quality is not discussed at all. Similarly, for the other medications mentioned (lisinopril and metoprolol) are used to treat hypertonia and/or heart diseases. Yet, both conditions can also be linked with male infertility. This is also not discussed.

It is also not clear why the authors used a cut-off of p<0.100 for significance instead of the standard P<0.05. The figures presented are difficult to understand, most probably because of a lack of explanations and the lack of legends.

Finally, the list of references is quite short for such a complex topic and is citing articles incorrectly.

Therefore, I was a bit two-minded for the rating. Although I ticked 'reject', I feel that due to the importance of the topic, the authors should be given an opportunity to better the manuscript. Therefore, I think a better option would be 'reject with an option to resubmit after complete rework'.

For more details of my review, I attached an anonymized PDF with my comments and corrections.

Reviewer #3: Dear Author

Thank you for your valuable manuscript.

1: Since most of the groups who refer for fertility are young and middle-aged people, it seems that the age limit should have been taken into account more seriously in the study.

2: History of surgery and duration of medication are also important, which are not well considered in the study.

3:Has it been asked about the use of fertility drugs in the history?

4:Is the information you need in the study in this long period of time of the studied subjects, in the patients' files in the same way?

**Do you want your identity to be public for this peer review?** For information about this choice, including consent withdrawal, please see our Privacy Policy

Reviewer #1: No

Reviewer #2: No

Reviewer #3: No

While revising your submission, please upload your figure files to the Preflight Analysis and Conversion Engine (PACE) digital diagnostic tool, https://://pacev2.apexcovantage.com/ . PACE helps ensure that figures meet PLOS requirements. To use PACE, you must first register as a user. Registration is free. Then, login and navigate to the UPLOAD tab, where you will find detailed instructions on how to use the tool. If you encounter any issues or have any questions when using PACE, please email PLOS at figures@plos.org . Please note that Supporting Information files do not need this step.

---

## [Author Response · Author response to Decision Letter 1]

19 Mar 2025

Dear, in the attached file (PDF) reviewer comments were shown including our reactions. other comments were answered as shown below:

• Why have the other comparisons (< 35 vs >+35 yrs, and <35 vs >=40 yrs) not been described? Was age as confounding factor considered for this analysis?

We thank the reviewer for asking this question and are happy to provide clarification. Figure 2 is an exploratory figure without corrections for confounding. In figure 2c we show LSQ prevalence for 5-year age categories concerning the total study group, whereas figure 2d is based on other cut-offs. As - based on figure 2c - LSQ prevalence does not increase up to 35 years, we chose <35 as reference category to which several classifications could be compared. We chose >=35, >=40, and >=45 based on methodological reasons. Higher age cut-offs would result in lower samples sizes decreasing reliability of results. The purpose of figure 2d was to show that the association of age with LSQ is stronger with increasing age cut-off.

We clarified this in the text.

• These AUCs, although significant, are merely indicating a predictive value around 50%. What were the classification criteria?

We acknowledge this question and think that comparison of AUCs does not add much value to the manuscript. We have deleted this section.

Low semen quality per ATC code

“Prevalence and association

LSQ prevalences with 95%CIs per ATC level are visualized in figures 3, 4 and 5, and plotted against the overall

prevalence of 52.8% indicated by the horizontal dotted line. LSQ prevalences were classified into 4 categories: 1]….”

• The number 52.8%:

We are happy to clarify this value. This percentage is the overall LSQ prevalence concerning the total study group. This percentages is shown in figure 2a, by the black point shape (all: n=7273). This overall LSQ prevalence was used as “background prevalence” to which subgroups were compared to explore subgroups with increased LSQ prevalence.

We clarified this in the text.

We hope we answered your questions sufficiently, and you will reconsider our manuscript for publication.

---

## [Editor Report · Decision Letter 1]

Thank you for submitting your manuscript to PLOS ONE. After careful consideration, we feel that it has merit but does not fully meet PLOS ONE’s publication criteria as it currently stands. Therefore, we invite you to submit a revised version of the manuscript that addresses the points raised during the review process.

**ACADEMIC EDITOR:**

Please write your response in a point-wise manner in the response letter.

The references should be uniform. Please improve the reference section.

Please correct all punctuation and grammatical errors.

If applicable, we recommend that you deposit your laboratory protocols in protocols.io to enhance the reproducibility of your results. Protocols.io assigns your protocol its own identifier (DOI) so that it can be cited independently in the future. For instructions see: https://://journals.plos.org/plosone/s/submission-guidelines#loc-laboratory-protocols . Additionally, PLOS ONE offers an option for publishing peer-reviewed Lab Protocol articles, which describe protocols hosted on protocols.io. Read more information on sharing protocols at https://://plos.org/protocols?utm_medium=editorial-email&utm_source=authorletters&utm_campaign=protocols .

We look forward to receiving your revised manuscript.

Kind regards,

Mukhtiar Baig, Ph.D.

Academic Editor

PLOS ONE

**Journal Requirements:**

While revising your submission, please upload your figure files to the Preflight Analysis and Conversion Engine (PACE) digital diagnostic tool, https://://pacev2.apexcovantage.com/ . PACE helps ensure that figures meet PLOS requirements. To use PACE, you must first register as a user. Registration is free. Then, login and navigate to the UPLOAD tab, where you will find detailed instructions on how to use the tool. If you encounter any issues or have any questions when using PACE, please email PLOS at figures@plos.org

---

## [Author Response · Author response to Decision Letter 2]

16 May 2025

Dear reviewers, Many thanks for your valuable comments. We think it greatly improved our manuscript and are happy to resubmit our work. We hope you will reconsider our submission. Thanks in advance on behalf of all co-authors, Peter ter Horst

5. Review Comments to the Author

Reviewer #1: Well written. However, it is a Retrospective study. As required by the journal, you need to have the data available. The graphs are also not easy to understand. Otherwise discussion and references were acceptable.

Dear Reviewer, Thanks for your comments. We added the raw data as a separate file, and we completed the captures of graphs and tables. Hope this will do.

Reviewer #2: In this study, the authors investigated the impact of medication on semen quality as per WHO criteria in a total of 7438 patients, of which 722 patients were under medication at the time of semen analysis, at Isala Fertility Center. Men were categorized based on the type of medication used and no medication. Age and period of sexual abstinence were investigated as confounding factors. Based on the medication used (the authors highlight metformin, metoprolol and lisinopril), the relevant medical conditions were concluded. Results show some significant effects of the medication taken on semen quality independent from age.

Although this is a very important aspect in andrological diagnostics for which the authors should be commended, the manuscript has significant problems starting from the English which has to be improved. In addition, important explanations are missing (e.g. ATC-7 level). The medical conditions are not mentioned in the Materials and Methods section. The tables that the authors are referring to are missing. The Figures do not have a legend. Furthermore, the authors link poor semen quality with intake of metformin. However, since metformin is used to treat diabetes, the impact of diabetes on semen quality is not discussed at all. Similarly, for the other medications mentioned (lisinopril and metoprolol) are used to treat hypertonia and/or heart diseases. Yet, both conditions can also be linked with male infertility. This is also not discussed.

It is also not clear why the authors used a cut-off of p<0.100 for significance instead of the standard P<0.05. The figures presented are difficult to understand, most probably because of a lack of explanations and the lack of legends.

Finally, the list of references is quite short for such a complex topic and is citing articles incorrectly.

Dear reviewer, Thanks for your comments. In the PDF document, we resolved your questions and remarks, hope this will do. See last section of this document where we reacted to the main comments. More specific:

- Important explanations are missing (e.g. ATC-7 level): we added this to the text

- The medical conditions are not mentioned in the Materials and Methods section: we don’t know the medical conditions of the patients included, instead we used medication as a proxy for medical condition, see sub-heading “Proxy” in the Methods section.

- The tables that the authors are referring to are missing. The Figures do not have a legend: we fixed this issue.

- Furthermore, the authors link poor semen quality with intake of metformin. However, since metformin is used to treat diabetes, the impact of diabetes on semen quality is not discussed at all. Similarly, for the other medications mentioned (lisinopril and metoprolol) are used to treat hypertonia and/or heart diseases. Yet, both conditions can also be linked with male infertility : we added this to the discussion section.

- It is also not clear why the authors used a cut-off of p<0.100 for significance instead of the standard P<0.05: because of the explorative design of the study we choose to use p<0.1 in univariate modeling.

- Finally, the list of references is quite short for such a complex topic and is citing articles incorrectly: we fixed this issue.

Therefore, I was a bit two-minded for the rating. Although I ticked 'reject', I feel that due to the importance of the topic, the authors should be given an opportunity to better the manuscript. Therefore, I think a better option would be 'reject with an option to resubmit after complete rework'.

For more details of my review, I attached an anonymized PDF with my comments and corrections.

Reviewer #3: Dear Author

Thank you for your valuable manuscript.

1: Since most of the groups who refer for fertility are young and middle-aged people, it seems that the age limit should have been taken into account more seriously in the study.

2: History of surgery and duration of medication are also important, which are not well considered in the study.

3:Has it been asked about the use of fertility drugs in the history?

4:Is the information you need in the study in this long period of time of the studied subjects, in the patients' files in the same way?

Dear reviewer, thanks for your comments and remarks. In answer to you, the following:

- Since most of the groups who refer for fertility are young and middle-aged people, it seems that the age limit should have been taken into account more seriously in the study. We indeed considered this as a limitation, however, as inclusion was from age of 18 years, the men who want to conceive are included, as no potential candidates (<18 years) were included.

- History of surgery and duration of medication are also important, which are not well considered in the study. We agree, however we didn’t have the data, so we stated in the discussion explicitly that this was a hypothesis-generating study instead of eg causal relationships.

- Has it been asked about the use of fertility drugs in the history? : we didn’t, however our fertility department covers almost the entire part of the Northern of The Netherlands and we used first time visit of the department as entry.

- Is the information you need in the study in this long period of time of the studied subjects, in the patients' files in the same way?: we don’t understand your question, may be you could be more specific?

Comments from the PDF, see also attached file:

Typo’s, grammar, references We changed typos, grammar issues and references as suggested throughout the text and in the reference section.

Tables and figures We completed the list of tables and figures, placed tables in the text, and added titles for all figures.

Abstract line 1 not only semen volume, sperm concentration and motility but also sperm function in general We changed as proposed

Abstract 1st paragraph provide more details in the types of medication e.g. medication classes, groups or specific medication/drugs We changed the text

Abstract 2nd paragraph ATC-7 code needs explanation We provided this in the text

why not the standard P-value for significance of P<0.05? In univariate analysis borderline significance can be used in modeling

Key points a clear take-home message indicating that medication can negatively affect semen quality needs to be provided, not only a hypothesis We added medication can negatively affect semen quality

Introduction, 1st paragraph please refer to the WHO definition of not being able to achieve a pregnancy after one year of unprotected sexual intercourse.

In fact, this is here a duplication of the information provided in the 3rd sentence. Hence, this sentence can be deleted.. We did and changed the text

We removed the duplication

Introduction 2nd paragraph This sentence needs rephrasing for better reading. At the moment, it is not a complete sentence. We rephrased the sentence

Methods, subsection potential confounders It might be better to have a separate section 'exclusion criteria'. Then, it will be clearer We added this to the study population section

Methods, subsection Proxy which other conditions? Please list This is just an example

Statistical methods this needs explanations. What do these levels describe? About ATC codes. We clarified this in the text

Start the stats section with the description of the software used. This includes the version of the software as well as the names of the suppliers with city and country. Thanks for the suggestion, we did.

Results The number of 21864 studies that were analyzed must be mentioned first. We did

Why have the other comparisons (< 35 vs >+35 yrs, and <35 vs >=40 yrs) not been described?

Was age as confounding factor considered for this analysis? We chose >=35, >=40, and >=45 based on methodological reasons. Higher age cut-offs would result in lower samples sizes decreasing reliability of results. We changed the text.

This is unclear what you are talking about.: “In many cases LSQ prevalences were high, but sample sizes were too small to reach (borderline) statistical significance, also indicated by wide 95%CIs. In some cases, the combination of prevalence and sample size resulted in a (borderline) statistically significant association, indicated by p-values in red (positive associations) and by pvalues in green (inverse association…” This is an explanation together with figures 3-5.

Discussion, 1st line namely?? Changed in: “We found that metoprolol, metformin, and lisinopril were associated with a high …”

Discussion, end of 1st paragraph Is this perhaps directly linked to the diabetes that these patients have? Diabetic patients are know to have impaired semen quality.

This must be analyzed and discussed. We added this to the discussion: “Finally, we found metformin was associated with LSQ. A review published bij Lotti and Maggi in 2022 showed that diabetes (type 1 and type 2) was associated with lower semen motility. [28] However, in our study it was hard to discriminate between medication effects (metformin) or (severity of) disease. Further studies are warranted because diabetes mellitus is one of the most common chronic conditions in Europe. At least 64 million adults and around 300 000 children and adolescents are estimated to be living with diabetes in the WHO European Region. [https://www.who.int/europe/news-room/fact-sheets/item/diabetes][29]

“

Discussion 2nd paragraph Metoprolol is used to treat high blood pressure and coronary heart disease. There is also a link between coronary hrt disease and male infertility (e.g. Farland et al., 2023; Skara et al., 2024). There are also indications that beta-blockers negatively affect male sexual function. We added this to the discussion:” We found a positive association between metoprolol and low semen quality, even when adjusted for age. The study by Guo et al showed that the intake of beta-blockers resulted in a lower sperm concentration, motility, and total count [20]. One of the proposed mechanisms by which beta-blockers may induce sexual dysfunction and low semen quality is by inhibiting the sympathetic nervous system, which is involved in the integration of erection, emission, and ejaculation, in the regulation of luteinizing-hormone secretion and the stimulation of release of testosterone [21]. A large cohorst study by Skara et al (2022) found

weake and no associations between infertility and cardiovascular disease (CVD) outcomes in men. This means a prescription of metoprolol is a proxy for CVD and therefor weaklyor notassociated with infertility.[21,22]

“

Calcium channel blockers such as Nifedipin can negatively affect sperm acrosome reaction. We added tthis to the text: “Nifedipine is known for its disruptive influence on sperm motility. A study by Kanwar (1993) showed that the pattern of motility changed within two hours from rapid and linear progression to slow or sluggish linear or non-linear movement and finally to non-progressive motility or even immotility. (in vitro) Scanning electron microscopic studies revealed disruptive changes in the head as well as tail region and coiling of spermatozoa after Nifedipine treatment. [30]

“

Some studies indicate that lisinopril at a low dose increased sperm count (Ghasi, 2012). We added this to the text:” In particular, lisinopril in low dose (2.5 mg daily), was used in a randomized trial to study sperm parameters. [27]Although the mean ejaculate volume was unchanged (P ≥ 0.093), the total sperm cell count and the percentage of motile sperm cells increased, whereas the percentage of sperm cells with abnormal morphology decreased. [27]Further studies are mandatory to explore the role of ACE inhibitors and semen quality.”

---

## [Editor Report · Decision Letter 2]

Medication, age and abstinence days associated with low semen quality: a cross-sectional study in more than 7000 men visiting the centre for reproductive medicine

PONE-D-25-03818R2

Dear Dr. Horst,

We’re pleased to inform you that your manuscript has been judged scientifically suitable for publication and will be formally accepted for publication once it meets all outstanding technical requirements.

Kind regards,

Mukhtiar Baig, Ph.D.

Academic Editor

PLOS ONE

---

## [Editor Report · Acceptance letter]

PONE-D-25-03818R2

PLOS ONE

Dear Dr. ter Horst,

I'm pleased to inform you that your manuscript has been deemed suitable for publication in PLOS ONE. Congratulations! Your manuscript is now being handed over to our production team.

Kind regards,

on behalf of

Professor Mukhtiar Baig

Academic Editor

PLOS ONE